# Conservative Management of Complicated Colonic Diverticulitis in Early and Late Elderly

**DOI:** 10.3390/medicina58010029

**Published:** 2021-12-24

**Authors:** Gennaro Perrone, Mario Giuffrida, Elena Bonati, Gabriele Luciano Petracca, Antonio Tarasconi, Gianluca Baiocchi, Fausto Catena

**Affiliations:** 1Department of Emergency Surgery, Parma University Hospital, 43100 Parma, Italy; glpetracca@gmail.com (G.L.P.); atarasconi@gmail.com (A.T.); 2General Surgery Clinic, Department of General Surgery, Parma University Hospital, 43100 Parma, Italy; mario.giuffrida4@gmail.com (M.G.); ebonati86@gmail.com (E.B.); 3Surgical Clinic, Department of Experimental and Clinical Sciences, University of Brescia, 25123 Brescia, Italy; gianluca.baiocchi@unibs.it; 4Department of Surgery, AAST Cremona, 26100 Cremona, Italy; 5General and Emergency Surgery, Bufalini Hospital, 47521 Cesena, Italy; faustocatena@gmail.com

**Keywords:** general surgery, acute diverticulitis, complicated diverticulitis, elderly, conservative treatment, conservative management

## Abstract

*Background and Objectives*: The management of complicated diverticulitis in the elderly can be a challenge and initial non-operative treatment remains controversial. In this study, we investigate the effectiveness of conservative treatment in elderly people after the first episode of complicated diverticulitis. *Materials and Methods*: This retrospective single-centre study describes 71 cases of elderly patients with complicated acute colonic diverticulitis treated with conservative management at Parma University Hospital from 1 January 2012 to 31 December 2019. Diverticulitis severity was staged according to WSES CT driven classification for acute diverticulitis. Patients was divided into two groups: early (65–74 yo) and late elderly (>75 yo). *Results*: We enrolled 71 elderly patients conservatively treated for complicated acute colonic diverticulitis, 25 males and 46 females. The mean age was 74.78 ± 6.8 years (range 65–92). Localized abdominal pain and fever were the most common symptoms reported in 34 cases (47.88%). Average white cells count was 10.04 ± 5.05 × 109/L in the early elderly group and 11.24 ± 7.89 in the late elderly group. CRP was elevated in 29 (78.3%) cases in early elderly and in 23 late elderly patients (67.6%). A CT scan of the abdomen was performed in every case (100%). Almost all patients were treated with bowel rest and antibiotics (95.7%). Average length of stay was 7.74 ± 7.1 days (range 1–48). Thirty-day hospital readmission and mortality were not reported. Average follow-up was 52.32 ± 31.8 months. During follow-up, home therapy was prescribed in 48 cases (67.6%). New episodes of acute diverticulitis were reported in 20 patients (28.1%), elevated WBC and chronic NSAID therapy were related to a higher risk of recurrence in early elderly patients (*p* < 0.05). Stage IIb-III with elevated WBC during first episode, had a higher recurrence rate compared to the other CT-stage (*p* = 0.006). *Conclusions*: The management of ACD in the elderly can be a challenge. Conservative treatment is safe and effective in older patients, avoiding unnecessary surgery that can lead to unexpected complications due to co-morbidities.

## 1. Introduction

Diverticulosis of the colon is an acquired condition that results from herniation of the mucosa through defects in the muscle layer [1].

Diverticulosis is defined as the presence of diverticula and is a common anatomical condition related to aging. Diverticulitis is a frequent elderly disease.

The prevalence of diverticulosis ranges from 32.6% in patients aged 50–59 years up to 71.4% in patients aged ≥80 years.

Conventionally, “elderly” has been defined as a chronological age of 65 years old or older, subjects from 65 through 74 years old are described as “early elderly” and those over 75 years old as “late elderly”.

Diverticular disease knowledge has evolved. Accumulation of visceral adipose tissue through chronic and low-grade inflammation are important in both pathogenesis of colonic diverticulosis and its complications. Many researchers have described the increase in complicated diverticulitis in metabolic syndrome, especially in patients with central obesity, dyslipidaemia and arterial hypertension. Additionally, NAFLD has been related to increased risk of complicated diverticulitis [2,3,4,5,6].

Colonic diverticulosis can present with a wide spectrum of clinical manifestations, ranging from asymptomatic disease to segmental colitis. Increased intraluminal pressure or thickened faecal material in the neck of diverticulum can lead to erosion of the luminal wall. Perforations can result in localized or extensive abscess, which can continue around the bowel wall and form a large inflammatory mass or extend to other organs. Free perforation into the peritoneum causes peritonitis [7].

Complicated diverticular disease is diagnosed in the presence of more severe clinical conditions, such as acute diverticulitis.

The treatment of complicated diverticulitis depends on the severity of the disease. Up to 50% of patients with acute complicated colonic diverticulitis (ACD) require emergency surgery [8,9,10].

Recurrences after medical treatment have been described in 13.3% to 36%, but only 3% to 5% develop complicated disease. In 38% of patients treated conservatively, persisting abdominal complaints have been reported during follow-up [11,12].

The primary endpoints of the study were to analyse the effectiveness of conservative treatment after the first episode of complicated diverticulitis in elderly patients and evaluate the recurrence rate and the need for surgery.

## 2. Methods

This retrospective single-centre population-based study describes 71 elderly patients with complicated acute left sided colonic diverticulitis (defined as diverticulitis with associated abscess, phlegmon, fistula, obstruction, bleeding or perforation) treated at Emergency Surgery Department of Parma University Hospital from 1 January 2012 until 31 December 2019, with institutional review board approval. An informed consent was obtained from all participants. Minimum one-year follow-up data were collected.

After first episode of acute complicated diverticulitis of the colon, we followed patients evaluating the onset and the severity of new episodes, the home therapy and the performed diagnostic tests.

Recurrent diverticulitis has been conventionally defined as the onset of abdominal symptoms after the first episode of diverticulitis.

New episodes that did not require hospitalization were considered uncomplicated diverticulitis, while cases confirmed with CT findings were considered complicated diverticulitis, according to WSES CT driven classification for acute diverticulitis [13] Table 1.

Inclusion criteria: Patients over 65, with a first episode of complicated acute left sided colonic diverticulitis, diagnosed with CT scan of the abdomen with intravenous contrast and treated with conservative management.

Exclusion criteria: Patients under 65; patients with uncomplicated acute diverticulitis; patients with non-colonic diverticulitis; patient with non-left sided colonic diverticulitis; patients with previous episodes of complicated acute diverticulitis; patients treated firstly with surgery.

We divided patients into two main groups: early elderly (subjects from 65 to 74 years old) and late elderly (subjects over 75 years old).

We analysed the following characteristics: age, sex, symptoms, personal history of diverticulosis documented with endoscopy or other diagnostic tests, comorbidity, immunocompetence status (history of recent cancer, chemotherapy treatment, immunological diseases, immunotherapy), BMI, chronic NSAID therapy excluding 5-ASA therapy (more than three times a week for more than three months in the last two years before acute diverticulitis onset), days of hospitalization, biochemical tests (wbc, hb, platelet, CRP, PCT, INR), diagnostic tests, CT classification, localization, treatment, and follow-up (new episodes, eventual surgical treatment, home therapy and follow-up tests) [14].

All patients were discharged after the improvement of clinical condition, biochemical tests and after the start of oral feeding.

Data analysis were performed using IBM SPSS Statistics by a biomedical statistician. Univariate and multivariate analysis were performed.

Statistical analysis was obtained for the main descriptive indexes. Position, dispersion and shape indices were calculated, among which, mean, median, mode, 5% trimmed mean, variance, standard deviation, interquartile range, minimum, maximum, asymmetry, and kurtosis. When relevant, standard errors and the corresponding 95% confidence intervals were also calculated.

Quantitative data are expressed as mean ± standard deviation (SD). The qualitative data were elaborated as absolute frequencies, relative frequencies, cumulated frequencies and percentages.

Categorical variables were evaluated by applying the Chi square test, while the numerical variables were evaluated by applying Mann–Whitney test.

All factors deemed to be statically significant for a *p*-value of less than 5% (*p* < 0.05).

The study has been approved by our institution independent ethics committee (Comitato etico AVEN–area vasta Emilia nord).

## 3. Results

In the present study, 71 elderly patients, treated firstly with conservative management of complicated acute colonic diverticulitis, were enrolled; 25 were males and 46 were females.

The ratio of male to female was 1.82. The mean age was 74.78 ± 6.8 years (range 65–92). A total of 37 patients were included in the early elderly group (52.11%) and 34 (47.89%) in the late elderly group.

Localized abdominal pain and fever were the most common symptoms in 34 cases (47.88%). Localized abdominal pain alone was reported in 14 patients (19.7%).

Seven patients were overweight (BMI more than 30 kg/m^2^). A total of 37 patients had a previous diagnosed diverticulosis (52.11%). A total of 17 patients were immunocompromised (23.94%) and chronic NSAID use were found in 14 cases (19.7%).

Average white cells count was 10.04 ± 5.05 × 109/L (range 2.5–24.9) in early elderly group and 11.24 ± 7.89 (range 3.5–50.25) in late elderly group.

CRP (normal range 0.5–5 mg/L) was elevated in 29 (78.3%) cases in early elderly and in 23 late elderly patients (67.6%) (105.28 ± 77.1 vs. 112.4 ± 747, *p* > 0.05).

A CT scan of the abdomen with intravenous contrast was performed in every case; 71/71 (100%). A CT scan of the abdomen with intravenous contrast, plain abdominal X-ray and abdominal ultrasound were performed in 42 patients (59.1%). A CT scan with intravenous contrast alone was performed seven times (9.8%).

According to WSES CT guided classification of diverticulitis [13], we found: 21 patients with stage Ia (29.5%); 28 stage Ib (39.4%); 10 stage IIa (14.0%); 7 stage IIb (9.8%); 5 stage III (7.4%) Table 2.

Almost all patients were treated with bowel rest and antibiotics (95.7%), according to WSES 2016 consensus conference recommendations [14].

Bowel rest was usually administered to every patient for the first day and oral feeding was started on the second day.

CT or ultrasound guided percutaneous drainage of abscessed diverticulitis were performed in six patients (8.44%).

Only three patients (4.2%) were treated with bowel rest and total parenteral nutrition, due to multiple antibiotic allergies. Broad spectrum antibiotics (metronidazole and amoxicillin) were administered 51 times (71.8%), while 12 times amoxicillin and metronidazole were administered in combination with other antibiotics (16.9%). Treatment with the combination of metronidazole and amoxicillin was performed for 6 days. Considering Antibiotic therapy, difference in efficacy was not found among the two main groups (*p* > 0.05).

Average length of stay was 7.74 ± 7.1 days (range 1–48), 6.3 ± 3.8 days (range 1–25) in early elderly group and 9.2 ± 9.2 days (range 2–48) in late elderly group.

Every patient was discharged from hospital uneventful. Thirty-day hospital readmission was not reported and not even 30-day mortality.

Follow-up ranged from 12 months up to 108 months. Average follow-up was 52.32 ± 31.8 months. Seven patients died for other reasons during follow-up (9.8%).

During follow-up, home therapy was prescribed in 48 cases (67.6%), rifaximin in 32 cases (45%) and mesalamine in 14 patients (19.7%). Diagnostic tests were performed in 42 patients (59.14%), colonoscopy was performed in 36 cases (50.7%) and CT scan of the abdomen in four patients (5.6%).

New episodes of acute diverticulitis were reported in 20 patients (28.1%), 11 in early elderly (55%) and nine in late elderly (45%). Early recurrence (within a year from the first episode) was reported in seven cases (35%), all cases in early elderly. Late recurrence (>12 months) was reported in the other 13 patients (65%), four in early elderly patients (30.7%) and nine in late elderly patients (69.3%).

During follow-up, episodes of complicated diverticulitis were reported and confirmed with a CT scan of the abdomen, with intravenous contrast in four patients (5.63%), equally distributed between early and late elderly patients, 2 vs. 2 cases. All these patients required surgery during the second admission (100%), as shown in Table 3 and Table 4.

Obesity, immunosuppression, gender, CRP levels were not related to recurrence rate in the two main groups (*p* > 0.05).

Pairwise comparison showed a higher recurrence rate in stage IIb-III, with elevated WBC during the first episode, compared to the other CT-stage (*p* = 0.006).

No recurrence rate differences were noted among patients with or without home therapy (*p* > 0.05).

## 4. Discussion

Our results suggest the efficacy and safety of conservative treatment for ACD in elderly patients. Follow-up data reported that 20 patients (28.1%) had new episodes of diverticulitis and only four patients had new high severity episodes of ACD (5.6%), requiring surgery. 94.4% of the elderly patients included in our study were treated conservatively during a follow-up period of up to nine years. No differences were found in early and late elderly patients. Elevated WBC and chronic NSAID therapy only were related to a higher risk of recurrence in early elderly patients. Significant leucocytosis was not found in late elderly patients, probably due to a lower inflammatory response in elderly people.

Buchwald et al. reported a recurrence rate of about 28% after conservative treatment in patients with Hinchey I-II ACD, but among these, only 19% needed surgery treatment [15].

Gregersen et al. reported the failure of conservative treatment in 19–21% of cases. The need for surgery was reported in 10 studies, with a median of 1.3%. This result is stackable with our findings [16].

Broderick-Villa et al. evaluated 2366 patients with ACD, who had undergone to conservative treatment; only 13% had a recurrence. They also reported that older patients had a lower recurrence [17].

The choice of the proper treatment of elderly patients with ACD should be carefully evaluated. The decision either to operate or not can be more difficult in an older patient with an abscess from perforated diverticulitis. Elderly patients are frailer and often surgery is unnecessary, leading to unexpected complications due to co-morbidities.

Literature findings support the idea that conservative treatment is effective in elderly patients, avoiding surgery in those who have completely recovered from ACD.

Surgery should be performed in selected cases only. If a patient is stable, antibiotic treatment alone and bowel rest can be offered, avoiding the possible morbidity correlated to an emergency procedure, as suggested by ourselves and the literature findings [18,19,20,21].

The limitation of this study is represented primarily by its retrospective nature in a single institution. In addition, many patients had lower CT-grade diverticulitis (grade Ia-Ib) and this might be one of the explanations for the low recurrence rate reported. It is probable that cases of relapse that occur shortly after the first episode are actually continuations of the same, rather than true recurrences. Furthermore, the follow-up period was too variable in our study. A multicentre cohort study would provide strong evidence and reduce the risk of bias. The low number of patients included in the study also adds limitations to this study. 

## 5. Conclusions

The management of ACD in the elderly can be a challenge. Conservative treatment is safe and effective in older patients, avoiding unnecessary surgery that can lead to unexpected complications due to co-morbidities.

## Figures and Tables

**Table 1 medicina-58-00029-t001:** 2015 WSES driven classification of diverticulitis.

Stage	Description
Uncomplicated diverticulitis	Diverticula, thickening of the wall, increased density of the pericolic fat
Complicated diverticulitis	
1A	Pericolic air bubbles or little pericolic fluid without abscess
1B	Abscess ≤ 4 cm
2A	Abscess > 4 cm
2B	Distant air (>5 cm from inflamed bowel segment)
3	Diffuse fluid without distant free air (no hole in colon)
4	Diffuse fluid with distant free air (persistent hole in colon)

**Table 2 medicina-58-00029-t002:** Abdominal CT-scan stage according to 2015 WSES driven classification of diverticulitis.

Ct-Scan Stage	Early Elderly	Late Elderly	TOT
Ia	9 (24.3%)	12 (35.2%)	21 (29.5%)
Ib	17 (45.9%)	11 (32.3%)	28 (39.4%)
IIa	7 (18.9%)	3 (8.8%)	10 (14.0%)
IIb	3 (8.1%)	4 (11.7%)	7 (9.8%)
III	1 (2.7%)	4 (11.7%)	5 (7.04%)
Total	37 (100%)	34 (100%)	71

**Table 3 medicina-58-00029-t003:** Clinical characteristics of early elderly patients with and without recurrence.

Parameters	No Recurrence (*n* = 26)	N Recurrence (*n* = 11)	*p*-Value
Early Elderly age (<65 yo)	68.8 ± 3.05 (65–74)	70.2 ± 2.86 (66–74)	0.195
Gender			0.528
Male	10 (38.4%)	3 (27.2%)	
Female	16 (61.5%)	8 (72.8%)	
Immunological status			0.832
Immunocompromised Patients	3 (11.5%)	1 (9%)	
Immunocompetent Patients	23 (88.4%)	10 (81%)	
BMI			0.152
BMI > 30 (kg/m^2^)	1 (3.8%)	2 (18.1%)	
BMI <30 (kg/m^2^)	25 (96.2%)	9 (81.9%)	
NSAID Therapy			0.022
Chronic NSAID therapy	3 (11.5%)	5 (45.4%)	
No Chronic NSAID therapy	23 (88.4%)	6 (54.6%)	
WBC	8.6 ± 3.5 (3.95–16.7)	13.32 ± 6.3 (2.5–24.9)	0.008
CRP	87.86 ± 71.7 (6.8–237.9)	138.37 ± 79.6 (16.8–250)	0.094

**Table 4 medicina-58-00029-t004:** Clinical characteristics of late elderly patients with and without recurrence.

Parameters	No Recurrence (*n* = 25)	N Recurrence (*n* = 9)	*p*-Value
Late Elderly age (>65 yo)	81.2 ± 4.39 (75–92)	79.6 ± 3.5 (75–86)	0.354
Gender			0.081
Male	11 (44%)	1 (11.1%)	
Female	14 (56%)	8 (88.9%)	
Immunological status			0.262
Immunocompromised Patients	10 (40%)	2 (22.2%)	
Immunocompetent Patients	15 (60%)	7 (77.8%)	
BMI			0.946
BMI > 30 (kg/m^2^)	3 (12%)	1 (11.1%)	
BMI < 30 (kg/m^2^)	22 (88%)	8 (88.9%)	
NSAID Therapy			0.563
Chronic NSAID therapy	5 (20%)	1 (11.1%)	
No Chronic NSAID therapy	20 (80%)	8 (88.9%)	
WBC	12.26 ± 8.84 (4.43–50.2)	8.4 ± 3.1 (3.5–12.6)	0.216
CRP	125.5 ± 80.9 (9–250)	75.5 ± 37.05 (16.1–106)	0.164

No differences were found on complicated recurrences among patients with early and late recurrence (*p* > 0.05).

## Data Availability

The original dataset generated during the current study is available from the corresponding author on reasonable request.

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
