# Peer review of "Conservative Management of Complicated Colonic Diverticulitis in Early and Late Elderly"

_medicina, 2021, doi:10.3390/medicina58010029_

Round 1
Reviewer 1 Report
This is an interesting paper of complicated diverticulosis in elderly. Author shows that complicated diverticulitis is common in this group of patients. Use of NSAIDs and leucocytosis are predictive of recurrence, and interestingly less than 50% of these patients have fever. Not surprisingly, significant leucocytosis was not found, as elderly often do not mount significant inflammatory response.
In my opinion the paper is interesting, relatively well written and I would recommend the following changes:
- In introduction references 1 and 2 are obsolete. There is new literature on diverticular disease ( diverticulosis and diverticulitis) in elderly and references should be updated: https://www.ncbi.nlm.nih.gov/pmc/articles/PMC6354172/
- Furthermore, a paragraph in introduction section is necessary to describe our evolving knowledge about diverticular disease. It is now believed to be metabolic disease, that share similar if not identical risk factors to NAFLD, metabolic syndrome etc ( https://pubmed.ncbi.nlm.nih.gov/33951119/)
- Line 70- this definition is not correct, please see above references and change accordingly
- It is interesting that only 7 patients were with BMI above 30. It would be interesting to report if BMI was associated with recurrence of the disease.
- Authors mention that CT abdomen was obtain in all cases. Please be more specific- was it abdominal CT with intravenous contrast, rectal contrast or both. In some cases, subtle diverticular perforation is not seen on CT with IV contrast and rectal contrast is needed
- What was duration of bowel rest and what was duration of antibiotic therapy? it should be investigated if duration of antibiotics therapy was associated with recurrence.
- Line 204 should start with Limitation of the study....
- majority of your patients were Ia and I8 group and this might be one of the explanation for low recurrence rate
Author Response
- done
- specific paragraph added
- changed
- done (BMI>30: 12% no recurrence, 11.2% recurrence)
- intravenous contrast
-
No recurrence rate differences were noted among patients with or without home therapy (p>0.05).
- done
- reported
Reviewer 2 Report
The treatment for diverticular disease has changed over the past decade with a trend towards more non-operative therapy and even avoidance of antibiotic therapy in patients with documented diverticulitis. This is a retrospective review of 71 patients over 7 years, starting in 2012. Two groups were studied based on age. Comments below:
- Grammatical and wording errors
- Average LOS 7 days with discharge criteria and home recommendations varied
- 28.1% recurrence rate with the contribution of an elevated WBC and NSAID use contributed
- 5.6% required operative intervention and 6 patients required IR drainage
- Varied follow up period, ranging from 12 months (too little) to 108 months to make a prediction
Author Response
- modified
- yes it is reported, but this does not affect recurrance rate,
No recurrence rate differences were noted among patients with or without home therapy (p>0.05).
-
Elevated WBC and chronic NSAID therapy were related to a higher risk of recurrence in early elderly patients (p<0.05).
- 5.6% required surgery for complicated recurrence, CT or ultrasound guided percutaneous drainage of abscessed diverticulitis were performed in 6 patients (8.44%) at first episode of complicated diverticulitis
- It is reported as a bias of the study: Furthermore, follow-up period was too variable in our study.
Reviewer 3 Report
The article presents an original research regarding the outcomes of conservative management in the elderly with acute left colonic diverticulitis. The paper is interesting and well written.
However, there are some issues that may be improved:
- Results section: The information provided in lines 126-141 may be easier to follow if presented in a table.
- in the Discussion section, it would be useful for the reader to specify when a conservative approach should not be indicated, what are the limits of a conservative management
General remarks:
the manuscript should be formatted according to the journal recommendations: font, size, abstract (non-structured), Conflict of interests, funding, and authors contribution at the end of the manuscript
Author Response
- done
- done
Round 2
Reviewer 1 Report
Authors have successfully responded to all of my questions
Reviewer 2 Report
Much improved manuscript. Again, not sure it adds to the current literature, but does highlight the diverticulitis treatment strategy at a single institution.
Reviewer 3 Report
The authors have revised the manuscript according to the recommendations.